# Is EBV Associated with Breast Cancer in Specific Geographic Locations?

**DOI:** 10.3390/cancers13040819

**Published:** 2021-02-16

**Authors:** Alison J. Sinclair, Manal H. Moalwi, Thomas Amoaten

**Affiliations:** School of Life Sciences, University of Sussex, Brighton BN1 9 QG, UK; mm2033@sussex.ac.uk (M.H.M.); Thomas.amoaten@nhs.net (T.A.)

**Keywords:** Epstein–Barr virus, breast cancer, gastric cancer, nasopharyngeal cancer, Burkitt’s lymphoma, geographic incidence

## Abstract

**Simple Summary:**

Epstein–Barr virus (EBV) is a virus that infects people and then remains within their bodies for life. Almost everyone is infected with EBV by the time they are an adult. EBV is called a cancer virus because it causes some cancers of blood cells and some cancers of the stomach and nose. Some scientists think that EBV may also cause some cases of breast cancer, but others disagree. It is difficult to be sure whether a pathogen that most people are infected with is the cause of any disease. Here, we review and discuss the evidence for and against the link between EBV and breast cancer and pose the questions that could help to answer whether EBV is a cause of breast cancer.

**Abstract:**

Epstein–Barr virus (EBV) is a virus that establishes a life-long infection in people, and infection with EBV is nearly ubiquitous by adulthood. EBV was identified from biopsy material from a child with Burkitt’s lymphoma (BL) in sub-Saharan Africa. EBV has a well-characterised role in the development of some cancers, notably, Burkitt’s lymphoma (BL), Hodgkin’s disease (HD), gastric carcinoma (GC), and nasopharyngeal carcinoma (NPC). Links have also been made between EBV and breast cancer (BC), but these have been controversial. For all EBV-associated cancers, the ubiquitous nature of infection with EBV, contrasted with the relatively rare development of cancer, highlights a problem of determining whether EBV is an aetiological agent of cancer. In addition, the geographic distributions of some EBV-associated cancers point to contributions from additional co-factors. Recent meta-analyses of the incidence of EBV within BC biopsies has revealed that the diversity in the conclusions remain, however, they also show more of an association between EBV and BC biopsies in some study locations. Here, we review the evidence linking EBV with BC, and conclude that the evidence for the presence of EBV in BC biopsies is concentrated in specific geographic regions but is currently insufficient to provide a causal link. We pose some questions that could help to resolve the question of whether EBV contributes to BC and probe the contribution EBV might make to the aetiology of BC.

## 1. Introduction

Epstein–Barr virus (EBV) is a human gamma herpes virus that the majority of adults are infected with worldwide [1,2,3,4]. It was isolated from a Burkitt’s lymphoma [5,6]. The virus is associated with a range of lymphomas: Burkitt’s lymphoma (BL), classical Hodgkin’s lymphoma (cHL), diffuse large B cell lymphoma (DLBCL), primary central nervous system lymphoma (PCNSL), primary effusion lymphoma (PEL), plasmablastic lymphoma [7,8], and natural killer T cell lymphomas (NKTL) [9]. EBV also is associated with carcinomas: nasopharyngeal carcinoma (NPC) [10,11,12,13] and gastric carcinoma (GC) [14,15,16]. In addition, EBV has been proposed to contribute to the development of breast cancer (BC), but this link is controversial. Examples include 0% association found in 107 BC samples from the USA using Epstein Barr virus encoded small RNAs in situ hybridization (EBER ISH) [17], this was also the case for six cases of lymphoepithelioma-like carcinomas of the breast from Europe [18] and 59 cases of BC from Europe [19]. Although 21% of BC samples from the UK scored positive for EBV DNA using a PCR assay, microdissection of the tumour cells showed that none contained the viral genome [20]. This contrasted with a 31% association with EBV found by PCR in 509 BC samples from Europe and North Africa, with a follow-up of 20 showing evidence of the EBV genome in some of the epithelial cells [21,22]. To understand why this question remains controversial and how this could be resolved, in this paper, we first consider the evidence for the association of EBV with other diseases and then compare this with the evidence for the association of EBV with BC.

### 1.1. Epstein–Barr Virus Lifecycle and B-Lymphocytes

EBV is transferred from person to person via saliva, normally resulting in an asymptomatic infection but leading to a hyperactive immune response termed infectious mononucleosis (glandular fever) in some cases [23]. During infection, EBV encounters quiescent primary B-lymphocytes and readily infects them, following interactions between the viral gp350 and cell surface CD21 receptor and between the viral glycoprotein 42 (gp42) with cellular human leukocyte antigen (HLA) [1]. Once the viral genome is internalised, the expression of a small sub-set of viral protein coding and regulatory RNA genes, a pattern known as latency III, drives the proliferation of the infected cells. EBV is categorised as a class I carcinogen on the basis of this potent ability to transform the growth of B-lymphocytes [24]. Some of the infected cells follow the natural pattern of differentiation into memory B-lymphocytes—they express fewer viral genes, a pattern known as latency 0, and the cells are thought to survive in a quiescent state for long periods. When the EBV-infected memory B-lymphocytes encounter their cognate antigen, the cells activate and differentiate into plasma cells. This drives EBV reactivation, leading to a cycle of viral lytic replication: the expression of the entire complement of viral genes, replication of the viral genome, and the extrusion of newly replicated virus particles [25]. These virions have the ability to either infect other cells in the body or to be transmitted to another person.

### 1.2. Epstein–Barr Virus and Epithelial Cells

EBV also infects some epithelial cells. This association was first identified in nasopharyngeal carcinoma (NPC) [26]. Investigations into the mechanism and consequences of EBV infection of epithelial cells revealed a distinct and more complex picture than the infection of B-lymphocytes. Infection of epithelial cells does not occur in the efficient manner that ensues for B-lymphocytes and the rapid and potent transformation that can be readily quantitated following B-lymphocyte infection has not been observed for epithelial cells [11]. Several more complex routes to infect epithelial cells have been described. These include B cell-mediated transfer [1,27], infection through cell-to-cell contact [28,29], and the direct infection of epithelial cells that express a recently identified viral receptor Ephrin A2 [30,31]. β1, and β5, -6, and -8 integrins have also been identified as cellular co-receptors, interacting with viral glycoprotein B (gB), glycoprotein H (gH), glycoprotein L (gL), and BamH1 Righwards reading frame 2 (BMRF2) proteins [1]. EBV lytic replication is readily detected in the more differentiated cells of oral hairy leukoplakia, an epithelial cell lesion found in immunosuppressed people [32]. This suggests that the stage of cell differentiation may control the expression of viral genes the expression of EBV lytic genes, with replication of the virus having been described as the default program for EBV in epithelial cells [11]. It is thought that the rarity of the occurrence of epithelial cell infection precludes the ready detection of EBV in epithelial cells in healthy people.

The EBV that most readily infects epithelial cells is most likely to have replicated in B-lymphocytes. This is because in B-lymphocytes, where HLA class II is expressed, the viral gp42 interacts with HLA class II in the endoplasmic reticulum, and this results in the degradation of much of the gp42. Virions with little or no gp42 are not very efficient at infecting B-lymphocytes where a gp42/HLA class II interaction aids infection, but the infection of epithelial cells is not compromised [1]. In contrast, EBV that has been replicated in epithelial cells, where HLA class II is not expressed, has high gp42 and is able to infect B-lymphocytes efficiently [1]. The recent identification of epithelial trophic isolates of EBV that are also more likely to enter the EBV lytic replication cycle demonstrates further complexity [33]. 

### 1.3. Established EBV-Associated Epithelial Cancers

The challenge of determining whether EBV causes any type of cancer is particularly difficult because of the life-long association of the virus with people. It is widely accepted that EBV has a role in at least two types of carcinoma: nasopharyngeal carcinoma (NPC) and gastric carcinoma (GC) [10,11,12,13]; in both, a sub-set of viral latency proteins, EBER RNAs and micro RNAs (mi-RNAs) (Latency II set), are expressed [14,15,16].

If we consider Koch’s postulates for assigning a pathogen as the cause of an infectious disease, then neither of these EBV-associated cancers would pass. To fulfil the criteria, the pathogen should be able to be isolated, transferred to a disease-free individual, and cause the same disease; this is clearly not the case for EBV because most people infected with the virus do not succumb to an EBV-associated cancer.

A revised view of whether a pathogen can be considered to cause a disease was proposed by Fredricks et al. [34] the first two criteria of which are relevant for this discussion:

“(i) A nucleic acid sequence belonging to a putative pathogen should be present in most cases of an infectious disease. Microbial nucleic acids should be found preferentially in those organs or gross anatomic sites known to be diseased (i.e., with anatomic, histologic, chemical, or clinical evidence of pathology) and not in those organs that lack pathology.

(ii) Fewer, or no, copy numbers of pathogen-associated nucleic acid sequences should occur in hosts or tissues without disease.”

Before EBV can be proposed as an aetiological agent for any cancer, evidence of the presence of the virus in the malignant tissue and not in samples without the disease is required. This is challenging for epithelial cancers because they often have a lymphocytic infiltrate that could contribute a source of latent EBV to a biopsy sample that is subject to extraction of nucleic acids followed by a gross analysis for viral genome or gene expression. However, in situ single cell analysis using histocytochemistry or in situ hybridisation (ISH) to detect viral gene expression can pin-point whether the EBV is associated with the malignant cells. Such an association has been established to be the case a proportion of NPC and GC cases [35,36].

Although EBV is not present in 100% of cases of either NPC and GC, further molecular understanding of these diseases has led to definitions of sub-types of each disease that are EBV-associated. For NPC, EBV is very strongly associated with two of the three types of NPC (described as non-keratinising or undifferentiated NPC) [37,38]. A strong geographic link between EBV-associated NPC with southern China, Southeast Asia, North Africa, and Greenland has been documented; however, the epidemiology underlying this geographic association has been described as enigmatic [39]. For GC, EBV is only associated with around 10% of GC cases. Although these cases have not been differentiated from the majority of GC cases on the basis of morphology, a molecular classification of GC now describes a separate sub-type for EBV-associated GC [40]. Therefore, for both NPC and GC, a clear definition for an association of the cancer with EBV was possible only after the sub-set of the cancers that are associated with EBV was specified.

A further question to ask when considering whether EBV infection may be a causal event for the development of the cancer is whether the virus infected a cell that then outgrew/survived its neighbours. This would suggest that infection with EBV predates and perhaps promotes a clonal expansion of cells. For NPC, current models suggest that EBV infects a pre-cancerous lesion of cells and further promotes cancer development [11,41] (Figure 1). In contrast, for GC, it is unknown whether EBV infects gastric cells that have sustained pre-existing malignant changes or whether it infects normal gastric epithelial cells. Irrespective of this, it is possible to ask whether EBV infection leads to a growth or survival advantage to the infected cells because EBV generates a unique molecular marker after it infects a cell, and this can be used to assess clonality of the virus infected cells within a tumour. The viral genome is present in a linear form in the virion and after it accesses the nucleus it circularises [42]. The junction is a repetitive region, and the join occurs in a manner that generates a random number of repeat units. This can be used as a bar-code to identify decedents of each infected cell. In a tumour biopsy, if a single bar-code predominates, it shows that the virus infected cells early in the disease and that these cells then formed a clonal outgrowth. In contrast, if no single bar-code predominates, then it is an indication that the virus infection was not as an early event driving the cancer development. Therefore, where EBV is clonal within the cancer biopsy, it provides a link between virus infection and the subsequent development of the cancer. Southern blot analysis revealed that both EBV-associated NPC and GC biopsies contain clonal EBV [43,44], suggesting that it drives the development of the cancer.

Nasopharyngeal epithelial cells are shown on the left. EBV is thought to infect precancerous cells that have undergone mutations and the viral genome is present in all cancer cells. Breast epithelia are shown on the right. EBV is able to infect normal breast epithelial cells, but although published evidence shows that that the virus is associated with some BC biopsies, the presence of viral genomes in epithelial cells is disputed and the presence of a viral genome in every cancer cell has not been reported.

### 1.4. The Potential of Epstein–Barr Virus in Breast Cancer

The potential of EBV to contribute to the development of BC is less clear than it is for NPC or GC. The presence of EBV genomes and expression of EBV genes in BC biopsies were first reported over 25 years ago [45], but a potential causal connection between EBV and BC continues to be contested [17,18,19,20,21,22]. Problems lie with the inclusion in biopsies of EBV-infected infiltrating lymphocytes and with the potential cross-reaction of EBV antibodies with off-target antigens [20]. A meta-analysis of early studies showed that EBV is present in BC biopsies more than it is in normal tissue (odds ratio of 6) [46]. However, this strong association did not differentiate between whether EBV is present in the malignant BC epithelial cells or in surrounding lymphocytes.

The use of in situ diagnostic approaches to discriminate between the association of EBV with malignant epithelial cells and infiltrating lymphocytes, together with two meta-analyses that included more recent studies, provided further evidence of a link between an association of EBV with BC (odds ratio (OR) 4.75, 95% confidence interval (CI) 2.53–8.92, *p* < 0.01) [47,48]. These analyses also point to a potential geographical difference in the frequency of an association of EBV with BC biopsies; specifically, a lower association with EBV was observed in samples from USA and western Europe compared to a higher association in samples from Asia [47] in one study and the highest frequency in South America in another [47,48].

However, the question as to whether EBV is present in malignant BC cells or in infiltrating lymphocytes does not yet have a broadly accepted answer. The most robust assay to identify individual cells detects an abundant gene product, EBER RNA, using in situ hybridisation (EBER-ISH) [49,50]. When considering studies that used this detection method to screen for the presence of EBV in malignant BC epithelial cells, we notice that a geographic difference is apparent. In the USA, a low to negligible association between EBV and malignant BC cells was reported [17,51], whereas in three neighbouring countries in northeast Africa and two countries in Asia, an association was reported between EBER expression and BC biopsies. The strength of the association ranged between 28 and 100% of the BC biopsies examined [52,53,54,55] (Table 1). However, as not all tumour cells in the biopsies were detected as EBER-positive, and some staining was present outside of the expected nuclear location, this does not mirror the pattern of EBER staining reported in GC and NPC. Therefore, questions remain about whether EBV is associated with malignant BC in these countries.

Considering the revisions to Koch’s postulates, we found some evidence that the EBV genome and EBV gene expression are found in BC biopsy samples more than in control samples, particularly in certain geographical locations. There is also published evidence that the virus is found in some of the malignant cells in these BC biopsies. Therefore, in these geographic regions, EBV cannot be ruled in or out in terms of fulfilling the postulates for an association with BC.

### 1.5. The Impact of EBV on Breast Epithelial Cells

The questions of whether and how EBV enters breast epithelial cells and whether EBV could contribute to disease development have been addressed in part. The evidence suggests that multiple routes may be available for EBV to infect breast epithelial cells. A recent study showed that direct infection by EBV is more efficient for primary breast epithelial cells and immortal mammary epithelial cell lines (HMLE, HMEC, and MCF10 A) than for transformed cell lines [57]. Furthermore, the authors show that the CD21 receptor plays a role in infection, but it does not account for all virus entry [57]. Whether the Ephrin A2 receptor plays a role in EBV infection of mammary epithelial cells remains to be determined.

The impact of the infection of the immortal HMEC cell line with EBV was an increase in the number and size of cellular 3D spheres that were formed [57]. These cells were immortalised following the addition of Tert and T-antigen, and an acceleration of BC formation was observed in the EBV-infected cells in a xenograft model when an activated Ras oncogene was also added to the infected cells and examined using in vitro growth and xenotransplantation assays [57]. The pattern of EBV genes that are expressed equate to those seen in EBV associated GC and NPC, a pattern known as “latency II” [57]. These include protein-coding genes, EBERS, and mi-RNAs. Furthermore, an altered gene expression pattern of host genes was associated with EBV-infection [57].

Other studies have shown that transformed BC cells lines can be infected with EBV when they are co-cultured with infected B cells, for example, the MDA-MB-231 cell line, and that infection with EBV can confer chemoresistance to the cells [58].

## 2. Conclusions and Remaining Questions

The question of whether EBV is associated with any cases of BC does not have a broadly accepted answer—much of the data in the meta-analyses rely on PCR and antibody detection, which each have potential caveats. With the “gold standard” EBER ISH assay, a geographically varied association of EBV with BC biopsies is seen. This suggests that any association of EBV with BC may be limited to specific geographic areas. This potential geographic association has parallels with the constrained geographic incidence of EBV-associated BL and EBV-associated NPC. From this, we suggest that remaining questions about the potential association of EBV with BC focus on cases in the regions reporting an association.

There is clear evidence that EBV is able to infect primary breast epithelial cells and to change their phenotype in vitro. However, answers to the question of whether EBV could be a driving force behind the development of BC or not requires further investigation.

Some burning questions that would enhance our understanding of the relevance of any EBV-association with BC are listed below.
What is the geographic extent of BC biopsies with evidence of an EBV association and are EBV genomes in epithelial cells?Is there a specific sub-type of EBV that is more common in BC biopsies associated with EBV?What is the molecular signature of viral gene expression in BC biopsies associated with EBV?Do epidemiological factors relate to the association with EBV in these regions?Is there is a molecular signature of cell gene mutations in BC biopsies associated with EBV in these regions?What is the molecular signature of the expression of cell genes in BC biopsies associated with EBV in these regions?

## Figures and Tables

**Figure 1 cancers-13-00819-f001:**
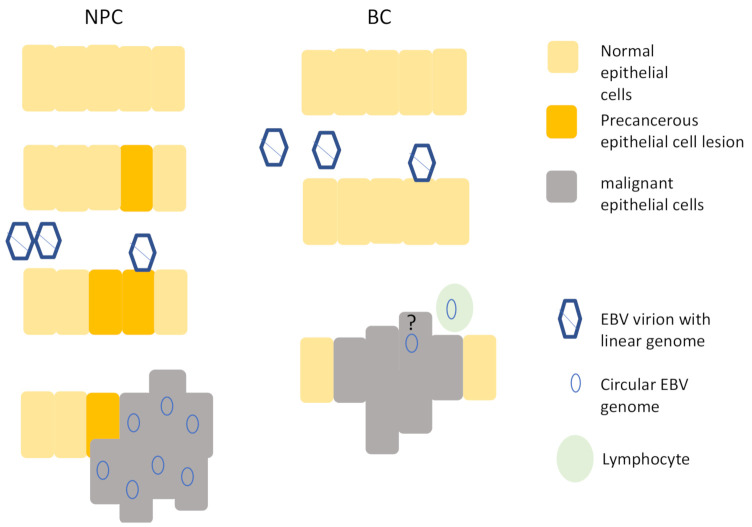
Model of Epstein–Barr virus (EBV) infection contrasting nasopharyngeal carcinoma (NPC) and breast cancer (BC). The question mark indicates the uncertainty in the field about whether the EBV detected in BC biopsies is present in epithelial cells or not.

**Table 1 cancers-13-00819-t001:** Reported association of breast cancer cells with EBV (as detected by EBER-ISH).

Frequency Positive (EBER-ISH)	Study Location	Citation
0%	USA	[17]
3%	USA	[56]
28%	Iraq	[55]
30%	India	[52]
36%	Eritrea	[53]
45%	Egypt	[55]
100%	Sudan	[54]

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
