# Peer review of "Is EBV Associated with Breast Cancer in Specific Geographic Locations?"

_cancers, 2021, doi:10.3390/cancers13040819_

Round 1
Reviewer 1 Report
The authors dealt with my comment appropriately.
Author Response
Reviewer 1
This is a review report that summarizes the linkage between EBV infection and breast cancer. The authors nicely and succinctly introduce us the pros and cons of the controversial issue, citing appropriate reports. I suggest only one modification before acceptance.
page 4 line 152; these opposing papers (ref 17-22) could be explained in more detail, one by one. This will clarify the point of argument even more.
Thank you for the response. We have expanded the detail of the opposing early papers [17-22] in the relevant section. This is included in new doc lines 47-52.
Reviewer 2 Report
Sinclair et al. have made minor changes to their manuscript. The major concerns addressed by this reviewer have not been tackled at all. Is this the right document? In the response there is reference to line numbers that do not match and the proposed changes are not clear in the revised document. Also, there is still no figure 2.
The main issue is still that none of the referenced articles show reasonable proof that EBV is present in breast carcinoma tumor cells. As stated before, the photographs on EBER ISH staining in these referenced articles are not convincing at all, if anything they actually strongly argue against the presence of EBV in breast cancer tumor cells. In general, not everything that has been published is true. Thus, the basis for the conclusions of Sinclair et al. is simply not there.
Author Response
Response to Reviewer 2 for cancers-1102761
There appears to have been confusion stemming from the wrong document being sent to the reviewer for re-review. We have made further changes in light of reviewer 2’s further comments, the revised response is below. The revised review text, a track-changes comparison with the original and a pdf file of the revised figure 1 are attached to the email send to the editorial office.
Reviewer 2
“This version better fits the response to the review comments. However, I still don't think that the comments have been incorporated well enough, as the paper still strongly implies that EBV+ breast carcinoma exists, for example as seen in the new graphical abstract. There is no clear evidence for that.
The changes in figure 1 are also clearly incorrect (EBV infecting normal breast epithelial cells would result in finding a 100% EBV+ tumor cells, not in a proportion of tumor cells)”
I present a summary of the changes and existing points that address the reviewers concerns below:
The simple summary ends with: “… we pose the questions that could help to answer whether EBV is a cause of breast cancer”.
The abstract ends with:“… conclude that the evidence for the presence of EBV in BC biopsies is concentrated in specific geographic regions but is currently insufficient to provide a causal link.”
In the main text on line 177-179 we state:
“A meta-analysis of early studies showed that EBV is present in BC biopsies more than it is in normal tissue (odds ratio of 6) [46]. However, this strong association did not differentiate between whether EBV is present in the malignant BC epithelial cells or in surrounding lymphocytes”.
On line 190 we state:
“However, the question of whether EBV is present in malignant BC cells or in infiltrating lymphocytes does not yet have a broadly accepted answer”.
On lines 198-201 we state:
“However, as not all tumour cells in the biopsies were detected as EBER-positive, and some staining was present outside of the expected nuclear location, this does not mirror the pattern of EBER staining reported in GC and NPC. Therefore, questions remain about whether EBV is associated with malignant BC in these countries”.
The title of table 1 has been changed to: “Table 1. Reported association of Breast Cancer cells with EBV (as detected by EBER-ISH)”.
On line 205-209 we conclude:
“Considering the revisions to Koch’s postulates, there is some evidence that the EBV genome and EBV gene expression are found in BC biopsy samples more than in control samples, particularly in certain geographical locations. There is also published evidence that the virus is found in some of the malignant cells in these BC biopsies. Therefore, in these geographic regions EBV can not be ruled in or out in terms of fulfilling the postulates for an association with BC”.
On line 235-237 we state:
“The question of whether EBV is associated with any cases of BC does not have a broadly accepted answer - much of the data in the meta-analyses rely on PCR and antibody detection which each have potential caveats”.
We amended the first question (line 251-252) to read:
“What is the geographic extent of BC biopsies with evidence of an EBV association and are EBV genomes in epithelial cells?”
Regarding the figures
We have requested that the graphical abstract is removed.
Fig 1 has been amended to show a “?” in the epithelial cell and the presence of an EBV-infected lymphocyte. The figure legend reads “… the presence of viral genomes in epithelial cells is disputed and the presence of a viral genome in every cancer cell has not been reported”.
Relating to the figure, line 244-246 reads:
“There is clear evidence that EBV is able to infect primary breast epithelial cells and to change their phenotype in vitro. However, answers to the question of whether EBV could be a driving force behind the development of BC or not requires further investigation”.
With best wishes,
Alison
Round 2
Reviewer 2 Report
The authors have addressed this reviewer's concerns adequately.
This manuscript is a resubmission of an earlier submission. The following is a list of the peer review reports and author responses from that submission.
Round 1
Reviewer 1 Report
This is a review report that summarizes the linkage between EBV infection and breast cancer. The authors nicely and succinctly introduce us the pros and cons of the controversial issue, citing appropriate reports. I suggest only one modification before acceptance.
page 4 line 152; these opposing papers (ref 17-22) could be explained in more detail, one by one. This will clarify the point of argument even more.
Reviewer 2 Report
Sinclair et al review evidence on association of the Epstein Barr virus with breast carcinoma. Although the subject and controversies are described in detail, one very important aspect is not discussed. That is the reliability of the EBER in situ hybridisation. It is known that this technique for example can stain cytoplasm of plasmacells and cytoplasma of neuroendocrine cells in the gut. Real evidence for a role of EBV in cancers would be strong and consistent staining of ALL cancer cell nuclei, as is the case for nasopharyngeal and gastirc carcinoma, Burkitt and Hodgkin lymphoma. The reviewer (experienced hematopathologist) checked all the EBER figures in referred manuscripts with positive findings on EBER ISH: ref.56 shows only cytoplasmic staining, ref.53 shows aspecific dotlike staining, also outside nuclei, ref. 50 shows nuclear staining, BUT in only 10-80% of tumor cells, ref.51 only a very small percentage of tumor cell nuclei positive (at best) and ref 52: most of the positivity is in lymphocytes (perhaps plasmacells). So, in conclusion, the reviewer finds no reliable evidence that EBV infection induces breast carcinoma. (Minor: figure 1 is unclear, figure 2 is missing). If the authors want to pursue this paper, I would suggest to them to include this expect and conclude that until now there is still no reliable evidence for EBV in causing breast cancer.
Reviewer 3 Report
A very nice, well written and comprehensive review. The main strength of the manuscript is the comprehensivenes. EBV role has been extensively described in different cell types and models for EBV-driven cancerogenesis have been presented. Indeed, this analysis may represent a good attempt t prompt further studies on this argument up.
The Language is clear and correct. Despite complete, the message remained concise.
I cannot see, by contrast, significant weakness.
Minor observations:
- Latency types have been discussed. However, the possible influence of miRNA was not discussed at all.